# Diverse Ensemble Evolution:
# Curriculum Data-Model Marriage

**Tianyi Zhou, Shengjie Wang, Jeff A. Bilmes**
Depts. of Computer Science and Engineering, and Electrical and Computer Engineering
University of Washington, Seattle
{tianyizh, wangsj, bilmes}@uw.edu

## Abstract

We study a new method "Diverse Ensemble Evolution (DivE$^2$)" to train an ensemble of machine learning models that assigns data to models at each training epoch based on each model's current expertise and an intra- and inter-model diversity reward. DivE$^2$ schedules, over the course of training epochs, the relative importance of these characteristics; it starts by selecting easy samples for each model, and then gradually adjusts towards the models having specialized and complementary expertise on subsets of the training data, thereby encouraging high accuracy of the ensemble. We utilize an intra-model diversity term on data assigned to each model, and an inter-model diversity term on data assigned to pairs of models, to penalize both within-model and cross-model redundancy. We formulate the data-model marriage problem as a generalized bipartite matching, represented as submodular maximization subject to two matroid constraints. DivE$^2$ solves a sequence of continuous-combinatorial optimizations with slowly varying objectives and constraints. The combinatorial part handles the data-model marriage while the continuous part updates model parameters based on the assignments. In experiments, DivE$^2$ outperforms other ensemble training methods under a variety of model aggregation techniques, while also maintaining competitive efficiency.

## 1 Introduction

Ensemble methods [7, 57, 31, 8] are simple and powerful machine learning approaches to obtain improved performance by aggregating predictions (e.g., majority voting or weighted averaging) over multiple models. Over the past few decades, they have been widely applied, consistently yielding good results. For neural networks (NN) in particular, ensemble methods have shown their utility from the early 1980s [72, 28, 39, 10] to recent times [50, 27, 66, 20]. State-of-the-art results on many contemporary competitions/benchmarks are achieved via ensembles of deep neural networks (DNNs), e.g., ImageNet [17], SQuAD [55], and the Kaggle competitions (https://www.kaggle.com/). In addition to boosting state-of-the-art performance of collections of large models, ensembles of small and weak models can achieve performance comparable to much larger individual models, and this can be useful when machine resources are limited. Inference over an ensemble of models, moreover, can be easily parallelized even on a distributed machine.

A key reason for the success of ensemble methods is that the diversity among different models can reduce the variance of the combined predictions and improve generalization. Intuitively, diverse models tend to make mistakes on different samples in different ways (e.g., assigning largest probability to different wrong classes), so during majority voting or averaging, those different mistakes cancel each other out and the correct predictions can prevail. As neural networks grow larger in size and intricacy, their variance correspondingly increases, offering opportunity for reduction by a diverse ensemble of such networks.

Randomization is a widely-used technique to produce diverse ensembles. Classical ensemble methods such as random initialization [17, 63], random forests [31, 8] and Bagging [7, 19], encourage diversity by randomly initializing starting points/subspaces or resampling the training set for different models. Ensemble-like methods for DNNs, e.g., dropout [61] and swapout [59], implicitly train multiple diverse models by randomly dropping hidden units out during the training of a single model. Diversity can also be promoted by sequentially training multiple models to encourage a difference between the current and previously trained models, such as Boosting [57, 23, 50] and snapshot ensembles [32]. Such sequential methods, however, are hard to parallelize and can lead to long training times when applied to neural networks.

Despite the consensus that diversity is essential to ensemble training, there is little work explicitly encouraging and controlling diversity during ensemble model training. Most previous methods encourage diversity only implicitly, and are incapable of adjusting the amount of diversity precisely based on criterion determined during different learning stages, nor do they have an explicit diversity representation. Some methods implicitly encourage diversity during training, but they rely on learning rate scheduling (e.g., snapshot ensembles [32]) or end-to-end training of an additive combination of models (e.g., mixture of experts [33, 34, 58]) to promote diversity, which is hard to control and interpret.

Moreover, many existing ensemble training methods train all models in the ensemble on all samples in the training set by repeatedly iterating through it, so the training cost increases linearly with the number of models and number of samples. In such case, each model might waste much of its time on a large number of redundant or irrelevant samples that have already been learnt, and that might contribute nearly zero-valued gradients. The performance of an ensemble on each sample only depends on whether a subset of models (e.g., half for majority voting) makes a correct prediction, so it should be unnecessary to train each model on every sample.

In this paper, we aim to achieve an ensemble of models using explicitly encouraged diversity and focused expertise, i.e., each model is an expert in a sufficiently large local-region of the data space, and all the models together cover the entire space. We propose an efficient meta-algorithm "diverse ensemble evolution (DivE$^2$)", that "evolves" the ensemble adaptively by changing over stages both the diversity encouragement and each model's expertise, and this is based on information available during ensemble training. It does this encouraging both intra- and inter-model diversity. Each training stage is formulated as a hybrid continuous-combinatorial optimization. The combinatorial part solves a data-model-marriage assignment via submodular generalized bipartite matchings; the algorithm explicitly controls the diversity of the ensemble and the expertise of each model by assigning different subsets of the training data to different models. The continuous part trains each model's parameters using the assigned subset of data. At each stage, all the models may be updated in parallel after receiving their assigned data.

A similar approach to encourage inter-model diversity was used in[15] but there diversity of different models is achieved by encouraging diverse subsets of features and the goal was to cluster the features into potentially overlapping groups; here we are encouraging diverse subsets of samples to be assigned to models during the training process and we are matching data samples to models.

We apply DivE$^2$ to four benchmark datasets, and show that it improves over randomization-based ensemble training methods on a variety of approaches to aggregate ensemble models into a single prediction. Moreover, with model selection based ensemble aggregation (defined below), DivE$^2$ can quickly reach reasonably good ensemble performance after only a few learning stages even though each individual model has poor performance on the entire training set. Furthermore, DivE$^2$ exhibits competitive efficiency and good of model expertise interpretability, both of which can be important in DNN training.

## 2 Diverse Ensemble Evolution (DivE$^2$): Formulation

### 2.1 Data-Model Marriage

An ensemble of models can make an accurate prediction on a sample without requiring each model making accurate predictions on that sample [41, 26, 42]. Rather, it requires a subset of models to produce accurate predictions, and the remainder may err in different ways. Hence, rather than training each model on the entire training set, we may in theory assign a data subset to each model. Then, each sample is learned by a subset of models, and different models are trained on different subsets thereby avoiding common mistakes across models.

Consider a weighted bipartite graph (see Fig. 1), with the set of $n = |V|$ training samples $V$ on the left side, the set of $m = |U|$ models $U$ on the right side, and edges $E \triangleq \{(v_j, u_i)|v_j \in V, u_i \in U\}$

connecting all sample-model pairs with edge weights defined by the loss $\ell(v_j; w_i)$ of sample $v_j = (x_j, y_j)$ (where $x_j$ is the features and $y_j$ is the label(s)) on model $u_i$ (where model $u_i$ is parameterized by $w_i$). We wish to marry samples with models by selecting a subset of edges having overall small loss. We can express this as follows:

$$\max_{\{w_i\}_{i=1}^m} \max_{A \subseteq E} \sum_{(v_j, u_i) \in A} (\beta - \ell(v_j; w_i)), \qquad (1)$$

where $\beta - \ell(v_j; w_i)$ translates loss to reward (or accuracy), and $\beta$ is a constant larger than any per-sample loss on any model, i.e., $\beta \geq \ell(v_j; w_i), \forall i, j$ [1].

With no constraints, all edges are selected thus requiring all models to learn all samples. As mentioned above, for ensembles, every sample need only be learned by a few models, and thus, for any sample $v$, we may wish to limit the number of incident edges selected to be no greater than $k$. This can be achieved using partition matroid $\mathcal{M}_V = (E, \mathcal{I}_V)$, where $\mathcal{I}_V = (I_1, I_2, \ldots, I_n)$ and $I_i \subseteq E$. $\mathcal{I}_V$ contains all subsets of $E$ where no sample is incident to more than $k$ edges in any subset, i.e. $\mathcal{I}_V = \{A \subseteq E : |A \cap \delta(v)| \leq k, \forall v \in V\}$, where $\delta(v) \subseteq E$ is the set of edges incident to $v$ (likewise for $\delta(u), u \in U$). Therefore, as long as a selected subset $A \subseteq E$ satisfies the constraint ($A \in \mathcal{I}_V$), every sample is selected by at most $k$ models.

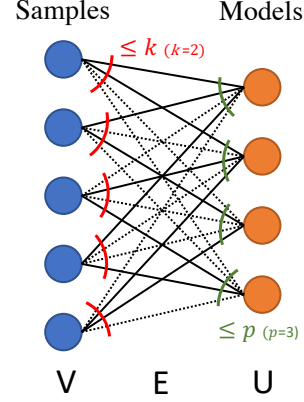

Samples      Models

$\leq k$ (k=2)

$\leq p$ (p=3)

V    E    U

Figure 1: Data-Model Marriage as Bipartite Matching.

With only the constraint $A \in \mathcal{I}_v$, different models can be assigned dramatically differently sized data subsets. In the extremely unbalanced case, $k$ models might get all the training data, while the other models get no data at all. This is obviously undesirable because $k$ models will learn from the same data (no diversity and no specialized and complementary expertise), while the others models learn nothing. Running time also is not improved since training time is linear in the size of the largest assigned data set, which is all of the data in this case. We therefore introduce a second partition matroid constraint $\mathcal{M}_U = (E, \mathcal{I}_U)$, which limits to $p$ the number of samples selected by each model. Specifically, $\mathcal{I}_u = \{A \subseteq E : |A \cap \delta(u)| \leq p, \forall u \in U\}$. Eq. (1) then becomes:

$$\max_{\{w_i\}_{i=1}^m} \max_{A \subseteq E, A \in \mathcal{I}_v \cap \mathcal{I}_u} \sum_{(v_j, u_i) \in A} (\beta - \ell(v_j; w_i)). \qquad (2)$$

The interplay between the two constraints (i.e., $\mathcal{I}_v$: $k$ models per sample, and $\mathcal{I}_u$: $p$ samples per model) is important to our later design of a curriculum that leads to a diverse and complementary ensemble. When $mp < nk$, $\mathcal{I}_u$ tends to saturate (i.e., $|A \cap \delta(u)| = p, \forall u \in U$) earlier than $\mathcal{I}_v$. Hence, each model generally has the opportunity to select the top-$p$ easiest samples (i.e., those having the smallest loss) for itself. We call this the "model selecting sample" (or early learning) phase, where each model quickly learns to perform well on a subset of data. On the other hand, when $mp > nk$, $\mathcal{I}_v$ tends to saturate earlier (i.e., $|A \cap \delta(v)| = k, \forall v \in V$), and each sample generally has the opportunity to select the best-$k$ models for itself. We call this the "sample selecting model" (or later learning) phase, where models may develop complementary expertise so that together they can perform accurately over the entire data space. We give conditions on which phase dominates in Lemma 1.

## 2.2   Inter-model & Intra-model Diversity

To encourage different multiple models to gain different proficiencies, the subsets of training data assigned to different models should be diverse. The two constraints introduced above are helpful to encourage diversity to a certain extent when $p$ and $k$ are small. For example, when $k = 1$ and $p \leq n/m$, no pairs of models share any training sample. Different training samples, however, can still be similar and thus redundant, and in this case the above approach might not encourage diversity when $p$ or $k$ is large. Therefore, we incorporate during training an explicit inter-model diversity term $F_{inter}(A) \triangleq \sum_{i,j \in [m], i < j} F\big((\delta(u_i) \cup \delta(u_j)) \cap A\big)$ and add it to the objective function in Eq. (2). This discourages model pairs from becoming too similar by discouraging their being assigned similar data. The set function $F : 2^E \rightarrow \mathbb{R}_+$ is chosen from the large expressive family of submodular functions, which naturally measure the diversity of a set of items [24]. A submodular function satisfies the diminishing return property: given a finite ground set $V$, any $A \subseteq B \subseteq V$ and an element

$v \notin B, v \in V$, we have $F(\{v\} \cup A) - F(A) \geq F(\{v\} \cup B) - F(B)$. Submodular functions have been applied to a variety of diversity-driven tasks to achieve good results [45, 44, 3, 54, 25, 22].

Another issue of Eq. (2) is that each model might select easy but redundant samples when constraint $\mathcal{I}_u$ dominates (the model-selecting-sample phase). This is problematic as each model might quickly focus on a small group of easy samples, and may overfit to such small region in the data space. We therefore introduce another set function $F_{intra}(A) = \sum_{i \in [m]} F'(\delta(u_i) \cap A)$ to promote the diversity of samples assigned to each model. Similar to $F$, we also choose $F'$ to be a submodular function. The optimization procedure now becomes:

$$\max_{W} \max_{A \subseteq E, A \in \mathcal{I}_v \cap \mathcal{I}_u} G(A, W) \triangleq \sum_{(v_j, u_i) \in A} (\beta - \ell(v_j; w_i)) + \gamma F_{inter}(A) + \lambda F_{intra}(A), \quad (3)$$

where $\gamma$ and $\lambda$ are two non-negative weights to control the trade-offs between the reward term and the diversity terms, and we denote $W \triangleq \{w_i\}_{i=1}^m$ for simplicity. By optimizing the objective $G(A, W)$, we explicitly encourage model diversity in the ensemble, while ensuring every sample gets learned by $k$ models so that the ensemble can generate correct predictions. A form of this objective has been called "submodular generalized matchings" [1] where it was used to associate peptides and spectra.

## 3 Diverse Ensemble Evolution (DivE$^2$): Algorithm

### 3.1 Solving a Continuous-Combinatorial Optimization

Eq. (3) is a hybrid optimization involving both a continuous variable $W$ and a discrete variable $A$. It degrades to maximization of a piecewise continuous function $H(W) \triangleq \max_{A \subseteq E, A \in \mathcal{I}_v \cap \mathcal{I}_u} G(A, W)$, with each piece defined by a fixed $A$ achieving the maximum of $G(A, W)$ in a local region of $W$. Suppose that $A$ is fixed, then maximizing $G(A, W)$ (or $H(W)$) consists of to $m$ independent continuous minimization problems, i.e., $\min_{w_i} \sum_{v_j \in V(A \cap \delta(u_i))} \ell(v_j; w_i), \forall i \in [m]$. Here $V(A) \subseteq V$ denotes the samples incident to the set of edges $A \subseteq E$, so $V(A \cap \delta(u_i))$ is the subset of samples assigned to model $u_i$. When loss $\ell(\cdot; w_i)$ is convex w.r.t. $w_i$ for every $i$, a global optimal solution to the above

---

**Algorithm 1** SELECTLEARN($k, p, \lambda, \gamma, \{w_i^0\}_{i=1}^m$)

1: **Input:** $\{v_j\}_{j=1}^n, \{l(\cdot; w_i)\}_{i=1}^m, \pi(\cdot; \eta)$
2: **Output:** $\{w_i\}_{i=1}^m$
3: **Initialize:** $w_i \leftarrow w_i^0 \ \forall i \in [m], t = 0$
4: **while** *not "converged"* **do**
5: $\quad W \leftarrow \{w_i^t\}_{i=1}^m$, define $G(\cdot, W)$ by $W$;
6: $\quad \hat{A} \leftarrow$ SUBMODULARMAX$(G(\cdot, W), k, p)$;
7: $\quad$ **if** $G(\hat{A}, W) > G(A, W)$ **then**
8: $\quad\quad A \leftarrow \hat{A}$;
9: $\quad$ **end if**
10: $\quad$ **for** $i \in \{1, \cdots, m\}$ **do**
11: $\quad\quad -\nabla \hat{H}(w_i^t) \leftarrow \frac{\partial}{\partial w_i^t} \sum_{v_j \in V(A \cap \delta(u_i))} \ell(v_j; w_i^t)$;
12: $\quad\quad w_i^{t+1} \leftarrow w_i^t + \pi \left( \{w_i^\tau, -\nabla \hat{H}(w_i^\tau)\}_{\tau \in [1,t]}; \eta^t \right)$;
13: $\quad$ **end for**
14: $\quad t \leftarrow t + 1$;
15: **end while**

---

continuous minimization can be obtained by various off-the-shelf algorithms. When $\ell(\cdot; w_i)$ is non-convex, e.g., each model is a deep neural networks, there also exist many practical and provable algorithms that can achieve a local optimal solution, say, by backpropagation.

Suppose we fix $W$, then maximizing $G(A, W)$ reduces to the data assignment problem (a generalized bipartite matching problem [43], see Appendix [71] Sec. 5.3 for more details), and the optimal $A$ defines one piece of $H(W)$ in the vicinity of $W$. Finding the optimal assignment is NP-hard since $G(\cdot, W)$ is a submodular function (a weighted sum of a modular and two submodular functions) and we wish to maximize over a feasibility constraint consisting of the intersection of two partition matroids ($\mathcal{I}_v \cap \mathcal{I}_u$). Thanks to submodularity, fast approximate algorithms [51, 48, 49] exist that find a good quality approximate optimal solution. Let $\hat{H}(W)$ denote the piecewise continuous function achieved when the discrete problem is solved approximately using submodular optimization, then we have $\hat{H}(W) \geq \alpha \cdot H(W)$ for every $W$, where $\alpha \in [0, 1]$ is the approximation factor.

Therefore, solving the max-max problem in Eq. (3) requires interaction between a combinatorial (submodular in specific) optimizer and a continuous (convex or non-convex) optimizer $\pi(\cdot; \eta)$ [2]. We alternate between the two optimizations while keeping the objective $G(A, W)$ non-decreasing.

Intuitively, we utilize the discrete optimization to select a better piece of $\hat{H}(W)$, and then apply the continuous optimization to find a better solution on that piece.

Details are given in Algorithm 1. For each iteration, we compute an approximate solution $\hat{A} \subseteq E$ using submodular maximization SUBMODULARMAX (line 6); in lines 7-9 we compare $\hat{A}$ with the old $A$ on $G(\cdot, W)$ and choose the better one; lines 10-13 run an optimizer $\pi(\cdot; \eta)$ to update each model $w_i$ according to its assigned data. Algorithm 1 always generates a non-decreasing (assuming $\pi(\cdot; \eta)$ does the same using, say, a line search) sequence of objective values. With a damped learning rate, only small adjustments get applied to $W$ and $G(\cdot, W)$. Thus, after a certain point the combinatorial part repeatedly selects the same $A$ (and line 7 eventually is always false), so the algorithm then converges as the continuous optimizer converges. [3]

## 3.2 Theoretical Perspectives

An interesting viewpoint of the max-max problem in Eq. (3) is its analogy to K-means problems [46]. Eq. (3) strictly generalizes the kmeans objective, by setting $\gamma = \lambda = 0, k = 1$, $p$ to be the number of desired clusters, and the loss to be the distance metric used in K-means (e.g., L2 distance), and the model to be a real valued vector of having the same dimension as $x$. Since K-means problem is NP-hard, our objective is also NP-hard.

We next analyze conditions for either of the constraints $(\mathcal{I}_v, \mathcal{I}_u)$ introduced in Section 2.1 to saturate. In the two extreme cases, we know that the "sample selecting model" constraint $\mathcal{I}_v$ saturates when $nk \ll mp$ (e.g., $k = 1$ and $p = n$), and the "model selecting sample" constraint $\mathcal{I}_u$ saturates when $nk \gg mp$ (e.g., $k = m$ and $p = 1$). However, it is not clear what exactly happens between them. The following Lemma shows the precise saturation conditions of the two constraints, with proof details in Section 5.1 of Appendix [71].

**Lemma 1.** *If SUBMODULARMAX is greedy algorithm or its variant, the data assignment $\hat{A}$ produced by lines 6-9 in Algorithm 1 fulfills: 1) $\mathcal{I}_v$ saturates, i.e., $|\hat{A} \cap \delta(v)| = k, \forall v \in V$, and $|\hat{A}| = nk$, if $k < {}^{mp+p}/_{n+(p-1)}$; 2) $\mathcal{I}_u$ saturates, i.e., $|\hat{A} \cap \delta(u)| = p, \forall u \in U$, and $|\hat{A}| = mp$, if $k > {}^{mp-p}/_{n-(p-1)}$; 3) when ${}^{mp+p}/_{n+(p-1)} \leq k \leq {}^{mp-p}/_{n-(p-1)}$, we have $|\hat{A}| \geq \min\{(k-1) + (m-k+1)p, (p-1) + (n-p+1)k\}$.*

As stated above, we can think the objective $H(W)$ as a piecewise function, where each piece is associated with a solution to the discrete optimization problem. Since it is NP-hard to optimize the discrete problem, Algorithm 1 optimizes $W$ on $\hat{H}(W)$, which is defined by the SUBMODULARMAX solutions, rather than on $H(W)$. Algorithm 1 has the following properties.

**Proposition 1.** *Algorithm 1: (1) generates a monotonically non-decreasing sequence of objective values $G(A; W)$ (assuming $\pi(\cdot; \eta)$ does the same) (2) converges to a stationary point on $\hat{H}(W)$; and (3) for any loss $\ell(u, w)$ that is $\beta$-strongly convex w.r.t. $w$, if SUBMODULARMAX has approximation factor $\alpha$, it converges to a local optimum $\hat{W} \in \arg\max_{W \in \mathcal{K}} \hat{H}(W)$ (i.e., $\hat{W}$ is optimal in an local area $\mathcal{K}$) such that for any local optimum $W_{loc}^* \in \mathcal{K}$ on the true objective $H(W)$, we have*

$$\hat{H}(\hat{W}) \geq \alpha H(W_{loc}^*) + \frac{\beta}{2} \cdot \min\{(k-1) + (m-k+1)p, (p-1) + (n-p+1)k\} \cdot \|\hat{W} - W_{loc}^*\|_2^2. \quad (4)$$

The proof is in Section 5.2 of Appendix [71]. The result in Eq. (4) implies that in any local area $\mathcal{K}$, if $\hat{W}$ is not close to $W_{loc}^*$ (i.e., $\|\hat{W} - W_{loc}^*\|^2$ is large), the algorithm can still achieve an objective $\hat{H}(\hat{W})$ close to $H(W_{loc}^*)$, which is a good approximate solution from the perspective of maximizing $G(A, W)$. Section 5.3 of Appendix [71] shows that the approximation factor is $\alpha = {}^1/_{2+\kappa_G}$ for the greedy algorithm, where $\kappa_G$ is the curvature of $G(\cdot, W)$. When the weights $\lambda$ and $\gamma$ are small, $\kappa_G$ decreases and $G(\cdot, W)$ becomes more modular. Therefore, the approximation ratio $\alpha$ increases and the lower bound in Eq. (4) improves. For general non-convex losses and models (e.g., DNNs), Eq. (4) degenerates to a weaker bound: $\hat{H}(\hat{W}) \geq \alpha H(W_{loc}^*)$.

## 3.3 Ensemble Evolution: Curricula for Diverse Ensembles with Complementary Expertise

For a model ensemble to produce correct predictions, we require only that every sample be learnt by a few (small $k$) models. Optimizing Eq. (3) with small $k$ from the beginning, however, might be harmful as the models are randomly initialized, and using the loss of such early stage models for the edge weights and small $k$ could lead to arbitrary samples being associated and subsequently

locked to models. We would, instead, rather have a larger $k$ and more use of the diversity terms at the beginning. To address this, we design an ensemble curriculum to guide the training process and to gradually approach our ultimate goal.

---
**Algorithm 2** Diverse Ensemble Evolution (DIVE$^2$)

---
1: **Input:** $\{(x_j, y_j)\}_{j=1}^n, \{w_i^0\}_{i=1}^m, \pi(\cdot; \eta), \mu, \Delta_k, \Delta_p, T$
2: **Output:** $\{w_i^t\}_{i=1}^m$
3: **Initialize:** $k \le m, p \ge 1$ s.t. $mp \le nk$,
$\qquad\qquad \lambda \in [0, 1], \gamma \in [0, 1]$
4: **for** $t \in \{1, \cdots, T\}$ **do**
5: $\quad \{w_i^t\}_{i=1}^m \leftarrow \text{SELECTLEARN}(k, p, \lambda, \gamma, \{w_i^{t-1}\}_{i=1}^m)$;
6: $\quad \lambda \leftarrow (1-\mu) \cdot \lambda, \gamma \leftarrow (1-\mu) \cdot \gamma$;
7: $\quad k \leftarrow \max\{\lceil k - \Delta_k \rceil, 1\}, p \leftarrow \min\{\lfloor p + \Delta_p \rfloor, n\}$;
8: **end for**

---

In Section 2.1, we discussed two ($mp < nk$ and $mp > nk$) extreme training regimes. In the first regime, there are plenty of samples to go around but models may only choose a limited set of samples, so this encourages different models to improve on samples they are already good at. In the first regime, however, inter-model diversity is important to encourage models to become sufficiently different from each other. Intra-diversity is also important in the first regime, since it discourages models from being trained on entirely redundant data. In the second regime, there are plenty of models to go around but samples may choose only a limited number of models. Each model is then given a set of samples that it is particularly good at, and further training further specialization. Since all samples are assigned models, this leads to complementary proficiencies covering the data space.

These observations suggest we start at the first regime $mp \le nk$ with small $p$ and large $k$, and gradually switch to the second regime with $mp \ge nk$ by slowly increasing $p$ and decreasing $k$. In earlier stages, we also should set the diversity weights $\lambda$ and $\gamma$ to be large, and then slowly reduce them as we move towards the second regime. It is worth noting that besides intra-model diversity regularization, increasing $p$ is also helpful to expand the expertise of each model since it encourages each model to select more diverse samples. Decreasing $k$ also helps to encourage inter-model diversity since it allows each sample to be shared by fewer models.

In later stages, the solution of Algorithm 1 becomes more exact. With $\lambda$ and $\gamma$ decreasing, according to Lemma 2, the curvature $\kappa_G$ of $G(\cdot, W)$ approaches 0, the approximation factor $\alpha = 1/2 + \kappa_G$ of greedy algorithm increases, and the approximate objective $\hat{H}(W) \ge \alpha H(W)$ becomes closer to the true objective $H(W)$. Moreover, during later stages when the "sample selecting model" constraint ($\mathcal{I}_v$) dominates and $\lambda$ and $\gamma$ are almost 0, the inner modular maximization is be exactly solved ($\alpha = 1$) and greedy algorithm degenerates to simple sorting.

The detailed diverse ensemble evolution (DivE$^2$) procedure is shown in Algorithm 2. The curriculum is composed of $T$ stages. Each stage uses SELECTLEARN (Algorithm 1) to (approximately) solve a continuous-combinatorial optimization in the form of Eq. (3) with pre-specified values of $(k, p, \lambda, \gamma)$ and initialization $\{w_i^{t-1}\}_{i=1}^m$ from the previous episode as a warm start (line 5). The procedure reduces $\lambda$ and $\gamma$ by a multiplicative factor $(1-\mu)$ in line 6, linearly decreases $k$ by $\Delta_k$ and additively increases $p$ by $\Delta_p$, in Line 7. Both $k$ and $p$ are restricted to be integers and within the legal ranges, i.e., $k \in [1, m]$ and $p \in [1, n]$. The warm start initialization is similar in spirit to continuation schemes used in previous curriculum learning (CL) [6, 5, 4, 35, 2, 60, 70] and SPL [40, 64, 62, 65], to avoid getting trapped in local minima and to stabilize optimization. As consecutive problems have the same form with similar parameters $(k, p, \lambda, \gamma)$, the solution to the previous problem might still evaluate well on the next one. Hence, instead of running lines 5-14 in Algorithm 1 until full convergence (as instructed by line 4), we run them for $\le 10$ iterations for reduced running time.

## 4 Experiments

We apply three different ensemble training methods to train ensembles of neural networks with different structures on four datasets, namely: (1) MobileNetV2 [56] on CIFAR10 [38]; (2) ResNet18 [29] on CIFAR100 [38]; (3) CNNs with two convolutional layers[4] on Fashion-MNIST ("Fashion" in all tables) [69]; (4) and lastly CNNs with six convolutional layers on STL10 [12][5]. The three training methods include DivE$^2$ and two widely used approaches as baselines, which are

- Bagging(BAG)[7]: sample a new training set of the same size as the original one (with replacement) for each model, and train it for several epochs on the sampled training set.

- RandINIT(RND): randomly initialize model weights of each model, and train it for several epochs on the whole training set.

Details can be found in Table 3 of Appendix [71]. We everywhere fix the number of models at $m = 10$, and use $\ell_2$ parameter regularization on $w$ with weight $1 \times 10^{-4}$. In DivE$^2$'s training phase, we start from $k = 6, p = {}^n/2m$ and linearly change to $k = 1, p = {}^{3n}/m$ in $T = 200$ episodes. We employ the "facility location" submodular function [14, 45] for both the intra and inter-model diversity, i.e., $F(A) = \sum_{v' \in V} \max_{v \in V(A)} \omega_{v,v'}$ where $\omega_{v,v'}$ represents the similarity between sample $v$ and $v'$. We utilize a Gaussian kernel for similarity using neural net features $z(v)$ for each $v$, i.e., $\omega_{v,v'} = \exp\left(-\|z(v) - z(v')\|^2/2\sigma^2\right)$, where $\sigma$ is the mean value of all the ${}^{n(n-1)}/2$ pairwise distances. For every dataset, we train a neural networks on a small random subset of training data (e.g., hundreds of samples) for one epoch, and use the inputs to the last fully connected layer as features $z$. These features are also used in the Top-$k$ DCS-KNN approach (below) to compute the pairwise $\ell_2$ distances to find the $K$ nearest neighbors.

## 4.1 Aggregation Methods using an Ensemble of Models

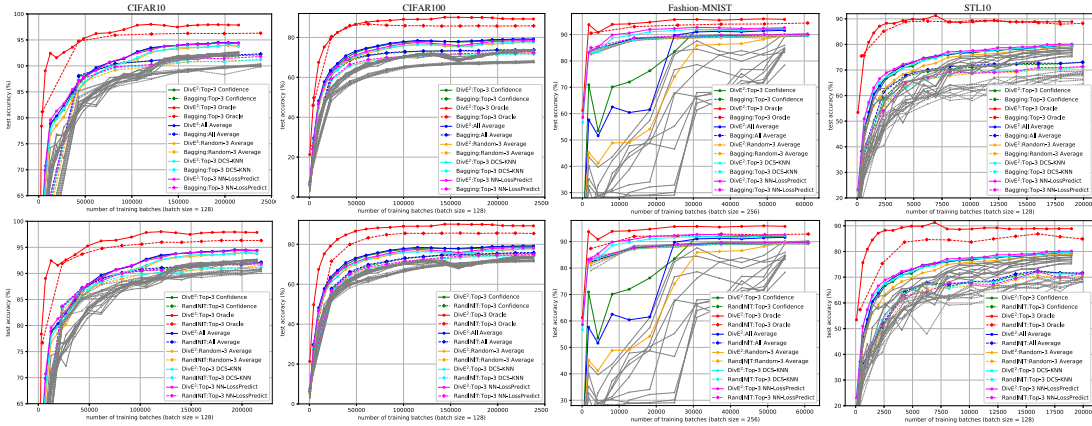

Figure 2: Compare DivE$^2$ with Bagging(upper row) and RandINIT(lower row) in terms of test accuracy (%) vs. number of training batches on CIFAR10, CIFAR100, Fashion-MNIST and STL10, with $m = 10$ and $k = 3$.

For ensemble model aggregation, when applying a trained ensemble of models to new samples, we must determine (1) which models to use, and (2) how to aggregate their outputs. Here we mainly discuss the first point about different model selection methods, because the aggregation we employ is either an evenly or a weighted average of the selected model outputs. Static model selection methods [72, 10, 53] choose a subset of models from the ensemble and apply it to all samples. By contrast, dynamic classifier selection (DCS) [11, 47, 73, 16] selects different subsets of models to be aggregated for each sample. KNN based DCS [68, 37] is a widely used method that usually achieves better performance than other DCS and static methods. When training, DivE$^2$ assigns different subsets of samples to different models, so for aggregation, we may benefit more from using sample-specific model selection methods. Therefore, we focus on DCS-type methods, in particular, the following:

- Top-$k$ Oracle: average the outputs (e.g., logits before applying softmax) of the top-$k$ models with the smallest loss on the given sample. It requires knowing the true label, and thus is a cheating method that cannot be applied in practice. However, it shows a useful upper bound on the other methods that select $k$ models for aggregation.
- All Average: evenly average the outputs of all $m$ models.
- Random-$k$ Average: randomly select $k$ models and average their outputs.
- Top-$k$ Confidence: select the top-$k$ models with the highest confidence (i.e., highest probability of the predicted class) on the given sample, and average their outputs.
- Top-$k$ DCS-KNN: apply an KNN based DCS method, i.e., find the $K$ nearest neighbors of the given sample from the training data, select the top-$k$ models assigned to the $K$ nearest neighbors by Top-$k$ Oracle, and average their outputs.
- Top-$k$ NN-LossPredict: train an L2-regression neural nets with $m$ outputs to predict the per-sample losses on the $m$ models by using a training set composed of all training samples and their losses on

the trained models. For aggregation, select the top-$k$ models with the smallest predicted losses on the given sample, and average their outputs.

We compare the three training methods used with the aforementioned aggregation methods with different $k$[6]. We summarize the highest test-set accuracy when $k = 3$ in Table 2, and show how the test accuracy improves as training proceeds (i.e., as the total training batches on all models increases) in Fig. 2. In Fig. 2, solid curves denote DivE$^2$, while dashed curves

Table 1: Total time (secs.) of DivE$^2$ and time only on SUBMODULARMAX.

| Dataset | CIFAR10 | CIFAR100 | Fashion | STL10 |
|---|---|---|---|---|
| Total time | 26790.75s | 34658.27s | 2922.89s | 4065.81s |
| SUBMODULARMAX | 1857.36s | 2697.36s | 81.64s | 378.84s |

denote the three baseline training methods. Different colors refer to different aggregation methods, and gray curves represent single model performance (gray solid curves denote models trained by DivE$^2$, while gray dashed curves denote models trained by other baselines). Similar results for $k = 5$ and $k = 7$ can be found in Appendix [71]. In addition, we also tested DivE$^2$ without the "model selecting sample" constraint and any diversity, which equals to [41, 26, 42] in multi-class case. It achieves a test accuracy of $90.11\%$ (vs. $94.36\%$ of DivE$^2$) on CIFAR10 and $71.01\%$ (vs. $78.89\%$ of DivE$^2$) on CIFAR100 when using Top-3 NN-LP for aggregation.

Top-$k$ Oracle (cheating) is always the best, and provides an upper bound. In addition, DivE$^2$ usually has higher upper bound than others, and thus has more potential for future improvement. Solid curves (DivE$^2$) are usually higher than dashed curves (other baselines) in later stages, no matter which aggregation method is used. Although diversity introduces more difficult samples and lead to slower convergence in early stages, it helps accelerate convergence in later stages. Although the test accuracy on single models achieved

Table 2: The highest test accuracy (%) achieved by different combinations of ensemble training and aggregation methods on four datasets, with $k = 3$. DivE$^2$ usually requires less training time than others to achieve the highest accuracy. The best non-cheating test accuracy (i.e., not Top-$k$ Oracle) is highlighted below.

| Train:Aggregation | CIFAR10 | CIFAR100 | Fashion | STL10 |
|---|---|---|---|---|
| BAG:Top-$k$ Oracle (Cheat) | 97.85 | 88.02 | 95.60 | 89.13 |
| BAG:All Average | 93.69 | 73.12 | 91.24 | 74.96 |
| BAG:Random-$k$ Avg. | 93.05 | 72.86 | 91.00 | 74.03 |
| BAG:Top-$k$ Confidence | 93.51 | 74.59 | 90.81 | 75.76 |
| BAG:Top-$k$ DCS-KNN | 92.86 | 73.06 | 91.39 | 74.07 |
| BAG:Top-$k$ NN-L.P. | 93.45 | 73.62 | 92.38 | 75.16 |
| RND:Top-$k$ Oracle (Cheat) | 97.80 | 87.01 | 95.71 | 89.54 |
| RND:All Average | 93.28 | 75.71 | 91.13 | 77.13 |
| RND:Random-$k$ Avg. | 93.11 | 75.56 | 90.77 | 76.75 |
| RND:Top-$k$ Confidence | 93.51 | 76.54 | 91.07 | 77.93 |
| RND:Top-$k$ DCS-KNN | 93.18 | 75.72 | 92.01 | 77.23 |
| RND:Top-$k$ NN-L.P. | 93.69 | 76.69 | 92.48 | 77.28 |
| DivE$^2$:Top-$k$ Oracle (Cheat) | 98.01 | 90.12 | 96.40 | 90.18 |
| DivE$^2$:All Average | 94.20 | **79.12** | 86.16 | 78.95 |
| DivE$^2$:Random-$k$ Avg. | 93.26 | 77.69 | 82.75 | 78.59 |
| DivE$^2$:Top-$k$ Confidence | 94.05 | 78.76 | 92.10 | 79.38 |
| DivE$^2$:Top-$k$ DCS-KNN | 93.81 | 77.61 | 92.10 | 79.23 |
| DivE$^2$:Top-$k$ NN-L.P. | **94.36** | 78.89 | **92.76** | **80.49** |

by DivE$^2$ is usually lower than those obtained by other baselines, the test accuracy on the ensemble is better. This indicates that different models indeed develop different local expertise. Hence, each model performs well good only in a local region but poorly elsewhere. However, their expertise is complementary, so the overall performance of the ensemble outperforms other baselines. We visualize the expertise of each model across different classes in Fig. 3 of Appendix [71] for Fashion-MNIST as an example. Among all aggregation methods, Top-$k$ NN-LossPredict and Top-$k$ DCS-KNN show comparable or better performance than other aggregation methods, but require much less aggregation costs when $k$ is small. As shown in Appendix [71], when changing $k$ from minority ($k = 3$) to majority ($k = 7$), the test accuracy of these two aggregation methods usually improves by a large margin. According to Table 1, DivE$^2$ only requires a few extra computational time for data assignment. The model training dominates the computations but is highly parallelizable since the updates on different models are independent.

**Acknowledgments**   This material is based upon work supported by the National Science Foundation under Grant No. IIS-1162606, the National Institutes of Health under award R01GM103544, and by a Google, a Microsoft, and an Intel research award. This research is also supported by the CONIX Research Center, one of six centers in JUMP, a Semiconductor Research Corporation (SRC) program sponsored by DARPA.

## Footnotes

[1]Although in theory the loss can be arbitrarily large, in practice, it is usually forced to be upper bounded by a constant for a stable gradient, e.g., a small $\sigma$ used in $-\log(p_i + \sigma) \leq -\log(\sigma)$ when computing cross entropy loss. Gradient clipping widely used in training neural nets also avoids arbitrarily large loss.

[2] The optimizer $\pi(\cdot; \eta)$ can be any gradient descent methods, e.g., SGD, momentum methods, Nesterov's accelerated gradient [52], Adagrad [18], Adam [36], etc. Here the first parameter $\cdot$ can include any historical solutions and gradients, and $\eta$ is a learning rate schedule (i.e., learning rate is $\eta^t$ for iteration $t$).

[3]Convergence is defined as the gradient $\nabla \hat{H}(W)$ w.r.t. $W$ being zero. In practice, we use $\|\nabla \hat{H}(W)\| \leq \epsilon$ for a small $\epsilon$.

[4]A variant of LeNet5 with 64 kernels for each convolutional layer.

[5]The network structure is from https://github.com/aaron-xichen/pytorch-playground.

[6]The $k$ used in aggregation fixed, and is different from the $k$ in training (which decreases from 6 to 1).

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
