[Supplementary Material · diverse_ensemble_evolution_supplementary.pdf]

# 5 Appendix

We define $\hat{A}_v \triangleq \hat{A} \cap \delta(v)$ (edges incident to sample $v$), and $\hat{A}^u \triangleq \hat{A} \cap \delta(u)$ (edges incident to model $u$) for simplicity.

## 5.1 Proof of Lemma 1

*Proof.* For the monotone non-decreasing objective $G(\cdot, W)$, greedy algorithm or its variant always tries to add more elements until some constraint(s) is violated. Hence, if the first constraint $\mathcal{I}_v$ does not saturate, i.e., there exists at least one $v' \in V$ such that $|\hat{A}_v| = k - x$ for some integer $1 \leq x \leq m$, and $|\hat{A}^u| = p, \forall u \in U \setminus U(\hat{A}_{v'})$, where $U(\hat{A}_{j'})$ represents the set of models incident to $\hat{A}_{v'}$. That is, for any model $u$ that sample $v'$ is not assigned to, the only reason that it cannot be assigned to sample $v'$ when $|\hat{A}_{v'}| < k$ is that the corresponding second constraint $\mathcal{I}_u$ already saturates. The number of models with saturated constraint $|\hat{A}^u| = p$ is $\left|U \setminus \hat{A}_{v'}\right| = m - k + x$. We then have

$$|\hat{A}| \geq |\hat{A}_{v'}| + \sum_{u \in U \setminus \hat{A}_{v'}} |\hat{A}^u| = (k - x) + (m - k + x)p \geq (k - 1) + (m - k + 1)p. \quad (5)$$

For the other $n - 1$ samples excluding sample $v'$, they need to satisfy the constraint $|\hat{A}_v| \leq k$, and they need to be assigned (with repetition) to the $(m - k + x)$ models such that each model gets $p$ samples, i.e.,

$$(n - 1)k \geq \sum_{v \in V \setminus v'} |\hat{A}_v| \geq \sum_{u \in U \setminus \hat{A}_{v'}} |\hat{A}^u| = (m - k + x)p \geq (m - k + 1)p. \quad (6)$$

Therefore, if the first constraint does not saturate, we must have

$$k \geq \frac{mp + p}{n + (p - 1)}. \quad (7)$$

An equivalent statement due to logical transposition rule is: if $k < {mp+p}/{n+(p-1)}$, the first constraint must saturate, and $|\hat{A}| = nk$. This completes the proof of the first statement in Lemma 1.

By following similar reasoning, if the second constraint does not saturate, i.e., there exists at least one $u' \in U$ such that $|\hat{A}^{u'}| = p - x$ for certain integer $1 \leq x \leq n$, we have

$$|\hat{A}| \geq |\hat{A}^{u'}| + \sum_{v \in V \setminus \hat{A}^{u'}} |\hat{A}_v| = (p - x) + (n - p + x)k \geq (p - 1) + (n - p + 1)k. \quad (8)$$

For the other $m - 1$ models excluding sample $u'$, they need to satisfy the constraint $|\hat{A}^u| \leq p$, and the $(n - p + x)$ samples need to be assigned (with repetition) to these $m - 1$ models such that each sample is assigned to $k$ models, i.e.,

$$(m - 1)p \geq \sum_{u \in U \setminus u'} |\hat{A}^u| \geq \sum_{v \in V \setminus \hat{A}^{u'}} |\hat{A}_v| = (n - p + x)k \geq (n - p + 1)k. \quad (9)$$

Hence, if the second constraint does not saturate, we must have

$$k \leq \frac{mp - p}{n - (p - 1)}. \quad (10)$$

Similarly, if $k > {mp-p}/{n-(p-1)}$, the second constraint must saturate, and $|\hat{A}| = mp$. This completes the proof of the second statement in Lemma 1.

By combing the conditions in Eq. (7) and Eq. (10), and the respective lower bounds of $|\hat{A}|$ in Eq. (5) and Eq. (8) under these two conditions, the third statement can be proved. □

## 5.2 Proof of Proposition 1

*Proof.* In each iteration, lines 7-9 in Algorithm 1 guarantees that $G(\hat{A}, W^t) \geq G(A, W^t)$ (where the superscript $t$ refers to the iteration index), and the gradient descent on $-\hat{H}(W^t)$ (or equally gradient ascent on $\hat{H}(W^t)$) guarantees that $H(W^{t+1}) \geq H(W^t)$, which implies $G(\hat{A}, W^{t+1}) \geq G(\hat{A}, W^t)$. Hence, we have $G(\hat{A}, W^{t+1}) \geq G(\hat{A}, W^t) \geq G(A, W^t)$, i.e., each iteration of Algorithm 1 does not decrease the objective $G(A, W)$ of the max-max problem in Eq. (3). So the algorithm generates

a monotonically non-decreasing sequence of objective values of $G(A, W)$. This completes the proof of the first statement.

By producing a monotonically non-decreasing sequence of objective values of $G(A, W)$, with a damped learning rate $\eta$, Algorithm 1 eventually stays on one piece of $\hat{H}(W)$ and converges to a stationary point on that piece with zero gradient. It will not end by oscillating amongst the non-differentiable boundaries between the pieces on $\hat{H}(W)$ because the algorithm can only visit each boundary point for at most one time due to the monotone non-decreasing objective values. This completes the proof of the second statement.

Given the data assignment $\hat{A}$ produced by lines 6-9 in Algorithm 1, the objective $G(\hat{A}, W)$ can be represented as the sum of $|\hat{A}|$ sample-wise loss functions in the following form.

$$G(\hat{A}, W) = \sum_{u_i \in U} \sum_{v_j \in V(\hat{A}^{u_i})} (\beta - \ell(v_j; w_i)) = \beta|\hat{A}| - \sum_{(v_j, u_i) \in \hat{A}} \ell(v_j; w_i). \tag{11}$$

Because each loss function $\ell(v_j; w_i)$ is $\beta$-strongly convex w.r.t. $w_i$, $-G(\hat{A}, W) = -\hat{H}(W)$ is $\beta|\hat{A}|$-strongly convex, which indicates that for any $W^*_{\text{loc}}$ and $\hat{W}$,

$$\hat{H}(\hat{W}) + \nabla\hat{H}(\hat{W})^T(W^*_{\text{loc}} - \hat{W}) - \hat{H}(W^*_{\text{loc}}) \geq \frac{\beta|\hat{A}|}{2}\|W^*_{\text{loc}} - \hat{W}\|_2^2. \tag{12}$$

Since $-\hat{H}(W)$ is $\beta|\hat{A}|$-strongly convex, any stationary point $\hat{W}$ achieved by Algorithm 1 is a local optimal solution within some local area $\mathcal{K}$. Hence, for any local optimal solution $W^*_{\text{loc}} \in \mathcal{K}$ on the true objective $H(W)$, the above inequality in Eq. (12) still holds. In addition, because $\nabla\hat{H}(\hat{W}) = \mathbf{0}$, Eq. (12) becomes

$$\hat{H}(\hat{W}) \geq \hat{H}(W^*_{\text{loc}}) + \frac{\beta|\hat{A}|}{2}\|W^*_{\text{loc}} - \hat{W}\|_2^2. \tag{13}$$

If SUBMODULARMAX has approximation factor $\alpha$, we further have $\hat{H}(W^*_{\text{loc}}) \geq \alpha \cdot H(W^*_{\text{loc}})$. Substituting this result and the lower bound for $|\hat{S}|$ from Lemma 1 into Eq. (13), we have

$$\hat{H}(\hat{W}) \geq \hat{H}(W^*_{\text{loc}}) + \frac{\beta|\hat{A}|}{2}\|W^*_{\text{loc}} - \hat{W}\|_2^2 \tag{14}$$

$$\geq \alpha H(W^*_{\text{loc}}) + \frac{\beta}{2} \cdot \min\{(k-1) + (m-k+1)p, (p-1) + (n-p+1)k\}\|\hat{W} - W^*_{\text{loc}}\|_2^2. \tag{15}$$

This completes the proof of the third statement. $\qquad\square$

## 5.3  Data Assignment as a Generalized Bipartite Matching Problem

The combinatorial optimization problem in Eq. (3) is a generalized bipartite matching problem [43] with monotone submodular evaluations and two matroid constraints, which is a special case of monotone submodular maximization with $p$-matroid constraint ($p = 2$). simple greedy algorithm can yield an approximation factor of $\alpha = 1/p+1$ [21]. This result can be further improved when the objective $G(\cdot, W)$ is close to modular. Specifically, $\alpha$ becomes $\alpha = 1/p+\kappa_G$ [13], which depends on the curvature $\kappa_G \in [0, 1]$ defined as

$$\kappa_G \triangleq 1 - \min_{j \in V} \frac{G(j|V\backslash j)}{G(j)}. \tag{16}$$

When $\kappa_G = 0$, $G(\cdot, W)$ is modular, and when $\kappa_G = 1$, $G(\cdot, W)$ is fully curved and the above bound recovers $\alpha = 1/p+1$. The objective $G(\cdot, W)$ in Eq. (3) is weighted sum of a modular function and two submodular functions. It becomes closer to modular as the weights $\lambda$ and $\gamma$ for the two submodular functions decrease, and $\kappa_G$ decreases accordingly. We therefore have the following Lemma:

**Lemma 2.** *Let $G(A) = M(A) + \lambda F(A)$ where $F(\cdot)$ is a monotone non-decreasing submodular function with curvature $\kappa_F$, $M(\cdot)$ is a non-negative modular function, and $\lambda \geq 0$. Then $\kappa_G \leq \frac{\kappa_F}{c/\lambda+1}$ where $c = \min_{j \in V} M(j)/F(j)$.*

*Proof.* We have

$$\kappa_G = 1 - \min_{j \in V} \frac{M(j) + \lambda F(j|V \setminus j)}{M(j) + \lambda F(j)} = \lambda \cdot \max_{j \in V} \frac{F(j) - F(j|V \setminus j)}{M(j) + \lambda F(j)}$$

$$= \lambda \cdot \max_{j \in V} \frac{1 - \frac{F(j|V \setminus j)}{F(j)}}{\frac{M(j)}{F(j)} + \lambda} \leq \frac{\lambda \cdot \kappa_F}{\min_{j \in V} \frac{M(j)}{F(j)} + \lambda} = \frac{\kappa_F}{c/\lambda + 1}$$

Where $c \triangleq \min_{j \in V} \frac{M(j)}{F(j)}$. □

In this paper, we apply the fast greedy procedure mentioned earlier [51, 48, 49] to the data assignment problem. It secures an approximation factor $\alpha = 1/2 + \kappa_G$, but might perform much better in practice.

## 5.4 Related Work

### 5.4.1 Three mostly used classical ensemble methods

Bagging [7]: bagging samples different training sets for different models before any training starts, and train all models in parallel, finally average all models' outputs as prediction. It does not adapt with the training process, i.e., the assignment of training data does not depend the performance of any model at any training stage. Multiple models can be trained in parallel so bagging is potentially applicable to deep neural nets, but might perform worse than simple average ensemble or dropout [41].

Boosting [57, 23, 50]: train a sequence of models one after another, and the weight of each training sample to train the next model depends on its classification accuracy achieved on previously trained models. It is adaptive and can build a strong ensemble model from multiple weak learners. However, it is not practical in training deep neural nets that usually require a long training time, because each model needs to wait the previous one to converge. In addition, boosting cannot adaptively adjust the training set during the training process of each model. The data assignment happens before any training begins.

Mixture of Experts (MoE) [33, 34]: they use a gating network to select a subset of models (experts) for each given sample. Because the gating network connects all the models together and forms a modular combination, which is usually a large neural network, end-to-end training is usually required, which is hard to parallelize and might result in expensive computations and heavy memory load. $\text{DivE}^2$ has similar idea of training experts, but is different from MoE methods in that 1) $\text{DivE}^2$ explicitly promotes diverse and complementary expertise on different experts; and 2) $\text{DivE}^2$ does not require end-to-end training (the gating network needs re-training if we remove or add models to the ensemble) and is able to train models in parallel, because each model is independently updated based on the assigned data in each learning stage.

### 5.4.2 Two mostly used ensemble methods for deep neural nets

Simple average might be the most widely used ensemble methods especially for deep neural nets. It trains different models on the same training set but initialize them randomly (and thereby promote diversity implicitly) at the beginning of optimization. It is simple to use but the diversity cannot be explicitly enforced. Moreover, it trains each model on the whole training set independently, so the training costs increase linearly with the number of models $m$.

Dropout [61]/Swapout [59]: implicitly gain an ensemble by randomly killing a portion of hidden nodes (i.e., set their outputs to be zeros) or skipping over layers (i.e., layer dropout). They implicitly average multiple models with different structures but with shared weights. They are different from explicitly training multiple models and explicitly enforcing the diversity between them. They can always be combined with other ensemble models including ours to further improve generation performance (in this case each model in the ensemble is implicitly an ensemble of models with different structures), and are orthogonal to methods explicitly training multiple models. ResNet [29] can also been explained as an ensemble of shallow models due to its shortcut link between nonconsecutive layers [67].

### 5.4.3 Two recently proposed ensemble methods for deep neural nets

Snapshot ensemble [32]: by using a cyclic learning rate scheduling, it can quickly converge to a local minimum, and escape from it by an increasing learning rate and then converge to the next local minimum. By repeating this process for several times, it can achieve multiple local minimum models.

The final ensemble is composed of the last several local minimum models. They can also be easily combined with other ensemble methods. One possible disadvantage of snapshot ensemble is that the computational cost to achieve so many local minimums (note the number could be much larger than the number of models used to compose the ensemble) can be very expensive, because it needs to sequentially get local minimum models one after another.

Sparsely gated mixture of experts [58]: it uses a parameterized gate to combine the outputs of all the models, and the gate is designed to only assign nonzero weights to a small number of models. This has been shown to be effective for some NLP tasks. The sparsity is helpful to develop diverse expertise on different models, but can easily cause imbalance loading problem in practice, as the extremely sparse weights given by the gate may always assign most data to few models. In addition, it needs to train thousands of models together with the gating network as a huge neural net in end-to-end manner, so the computational costs are very expensive, and it is not easy to synchronize the training process and accelerate it in parallel.

#### 5.4.4 Other Related Works

Model compression [9] or knowledge distillation [30] learns a single small model to imitate an ensemble of models, so the aggregation requires much less computation and memory. These methods mainly focus on improving the aggregation efficiency, and can also be applied to the diverse ensemble achieved by DivE$^2$.

### 5.5 Discussion

In DivE$^2$, the parameters $k, p, \lambda, \gamma$ defining the learning goal of each stage are gradually change according to a pre-defined schedule. In the future, we plan to use reinforcement learning to train an agent to adaptively select these parameters based some features representing the state of the current learning stage.

In DivE$^2$, we extend the concept of "machine teaching" to "machine education" by emphasizing the dynamic interaction between teacher (data assignment) and students (models) during the learning process. In machine education, the curriculum is composed of a sequence of learning goals for different learning stages, and each goal (i.e., an optimization in the form of Eq. (3)) is achieved based on the current performance of models and the data distribution. In contrast, machine teaching aims at finding the best "teaching set", which is the final goal of the teacher, but does not delicately optimize the learning process. While machine teaching might be useful to convex optimization, machine education that optimizes the learning process (i.e., the curriculum) is more suitable to non-convex optimization for training deep neural networks.

### 5.6 More Experiment Details

Figure 3: Test accuracy (%) per class on each single model from the ensemble trained by Bagging(left, after 18750 total training batches), RandINIT(middle, after 18750 total training batches) and DivE$^2$ (right, after 18249 total training batches) on Fashion-MNIST. This figure reflects the expertise of each model on different classes. Comparing to Bagging and RandINIT, the models learned by DivE$^2$ show diverse and complementary expertise.

Table 3: Details regarding the datasets.

| Dataset | CIFAR10 | CIFAR100 | Fashion | STL10 |
|---|---|---|---|---|
| #Training | 50000 | 50000 | 60000 | 5000 |
| #Test | 10000 | 10000 | 10000 | 8000 |
| #Feature | $3 \times 32 \times 32$ | $3 \times 32 \times 32$ | $28 \times 28$ | $3 \times 96 \times 96$ |
| #Class | 10 | 100 | 10 | 10 |

Figure 4: Compare DivE$^2$ with Bagging(left column) and RandINIT(right column) in terms of test accuracy (%) vs. number of training batches on CIFAR10, with $m = 10$ MobileNetV2 models trained, and using different $k$ values ($k = 3, 5, 7$ from top to bottom) for aggregation.

Figure 5: Compare DivE$^2$ with Bagging(left column) and RandINIT(right column) in terms of test accuracy (%) vs. number of training batches on CIFAR100, with $m = 10$ ResNet18 models trained, and using different $k$ values ($k = 3, 5, 7$ from top to bottom) for aggregation.

Figure 6: Compare DivE$^2$ with Bagging(left column) and RandINIT(right column) in terms of test accuracy (%) vs. number of training batches on Fashion-MNIST, with $m = 10$ modified LeNet5 models trained, and using different $k$ values ($k = 3, 5, 7$ from top to bottom) for aggregation.

Figure 7: Compare DivE$^2$ with Bagging(left column) and RandINIT(right column) in terms of test accuracy (%) vs. number of training batches on STL10, with $m = 10$ CNN models trained, and using different $k$ values ($k = 3, 5, 7$ from top to bottom) for aggregation.