[Reviews · NeurIPS 2018]

Reviewer 1



This paper proposes a new technique for training ensembles of predictors for supervised-learning tasks. Their main insight is to train individual members of the ensemble in a manner such that they specialize on different parts of the dataset reducing redundancy amongst members and better utilizing the capacity of the individual members. The hope is that ensembles formed out of such predictors will perform better than traditional ensembling techniques. The proposed technique explicitly enforces diversity in two ways: 1. inter-model diversity which makes individual models (predictors) different from each other and 2. intra-model diversity which makes predictors choose data points which are not all similar to each other so that they don't specialize in a very narrow region of the data distribution. This is posed as a bipartite graph matching problem which aims to find a matching between samples and models by selecting edges such that the smallest sum of edge costs is chosen (this is inverted to a maximization problem by subtracting from the highest constant cost one can have on the edges.) To avoid degenerate assignments another matching constraint is introduced which restricts the size of samples selected by each model as well. This will ensure that no model starves of samples. The two constraints are posed as matroid constraints on a utility which consists of three terms: 1. The sum of edge costs to be minimized (posed as a maximization) 2. a inter-model diversity term and 3. a intra-model diversity term. The last two terms are chosen from the family of submodular functions which are well-known to naturally model diversity and diminishing return properties. Since the overall objective function results in a combinatorial (submodular) and a continuous optimization problem the proposed solution method alternates between the two. First the discrete optimization is solved to find a better matching and then the continuous optimization is run to train the models on the new matches. A curriculum is designed for this process so that initially there is more weight on diversity terms and then reducing them slowly. Experiments are presented on CIFAR10, CIFAR100, Fashion and STL10 datasets. Bagging and random initialization of DNNs are used as the training baselines. A number of inference schemes are demonstrated with the best performing ones being the Top-K NN-LossPredict which trains a NN to predict for a given sample which m models are the best ones. On all four datasets DivE2 performs the best using mostly Top-K NN-LP as the inference method. On CIFAR-100 DivE2 + all average inference works the best. Comments: - The paper is generally well-written and easy to understand. Thanks!! - The proposed method is interesting, novel and tackles a relevant problem. - Major concerns: 1. Missing baselines: In addition to the bagging and random initialization training methods the natural other ones to try which try to incorporate diversity are [38] "Why M-heads are better than one..." and [not currently cited] "Stochastic MCL for Training Diverse Deep Ensembles, Lee et al.", NIPS 2016 and since the authors are not tackling structured prediction tasks but multi-class classification tasks: "Contextual Sequence Prediction with Application to Control Library Optimization" by Dey at al., RSS 2012 which also provides the theoretical groundings for [10]. 2. Datasets and tasks: The authors are currently focussed on small multi-class datasets. It will be much stronger statement to show results on larger datasets like ImageNet and/or MS COCO or other non-vision large datasets for multi-class tasks. (Note that most of the related work in this area have demonstrated on structured prediction tasks but I understand that the scope of this paper focused on multi-class prediction problems.) I am happy to be convinced why any/all of the above baselines are not suitable candidates for comparison to the proposed method. 3. List of relevant work which should be discussed and cited: "Stochastic MCL for Training Diverse Deep Ensembles", Lee et al., NIPS 2016 "Contextual Sequence Prediction with Application to Control Library Optimization" by Dey at al., RSS 2012 "Learning Policies for Contextual Submodular Prediction", Ross et al. ICML 2013. "Predicting Multiple Structured Visual Interpretations", Dey et al., ICCV 2015 "Confident Multiple Choice Learning", Lee et al., ICML 2017 - Minor concerns: 1. From the results presented in Table 2 it seems that Top-k NN-LP should always be used as the inference technique even with other training methods like BAG or RND. Is this assessment fair? 2. On CIFAR-100 all-average is performing the best although Top-K NN-LP is not that far behind. On a larger dataset like ImageNet or MS Coco will all average still do the best or do the authors think that Top-K NN-LP will prevail?

Reviewer 2



This paper introduces a new technique that trains a diverse ensemble by assigning hypotheses to samples. Its main benefit compared to traditional ensemble methods is the gradual specialization of experts through the addition of intra/inter-model diversity terms in the objective. While the notion of specialization has been studied previously, this specific formulation with its switching between two MOs ('model selecting sample' and 'sample selecting model') is new and is of significance to the ML community, and may have especially high impact for datasets with a large amount of sample heterogeneity (for instance, data collected across multiple populations). I find it a bit surprising that a boosting-type method (maybe gradient boosting) was not used in the comparison. These methods are known for allowing hypotheses to complement each others' strengths. According to Table 2, DivE performs clearly better in terms of magnitude on CIFAR100 and STL10, however, for CIFAR10 and Fashion the improvements are small - are these results statistically significant? Are there cases when Dive is expected to perform a lot better? Maybe the more complex the dataset, the higher the benefit? Also, it seems that the Top-k-NN-LP strategy performs better in general regardless of the ensemble type used, with the exception of Dive all avg. on CIFAR100. It may be worth investigating why this happens - can we draw any intuition about the relative merits of these scoring strategies. Clarity: Figure 2 is illegible unless magnified. As a table is also provided, it may be a good idea to select only two more representative (or interesting) examples to show here, with the rest relegated to the appendix. It may be an idea to show only the best for BAG, RND and DivE on the same plot for each dataset. Suggested reference: "Projection Retrieval for Classification" (NIPS 2012) presents a data partitioning ensemble method with a submodular objective. The implementation of this technique is highly non-trivial, so acceptance of the paper should be conditioned on making the code and the experiments publicly available. Update: The authors explained why they haven't compared against boosting in the feedback, so my main misgiving was addressed.

Reviewer 3



Contributions: This paper addresses the problem of training diverse ensembles, with an additional goal of also training more efficiently. To do this, they introduce the DivE2 algorithm, which solves a continuous-combinatorial constrained optimization problem. To enforce diversity, DivE2 introduces two matroid constraints with parameters p and k, where p controls the maximum number of examples selected by each model, and k controls the maximum number of models that may be assigned to a given example. DivE2 also introduces two submodular regularization terms: one controling inter-model diversity (diversity of examples assigned between pairs of models), and one controlling intra-model diversity (diversity of examples assigned to a given model). The strength of these regularization terms are controlled by lambda and gamma parameters. DivE2 evolves the diversity requirements over the course of T “episodes” of training by gradually trading off p and k, and also gradually decreasing lambda and gamma. Strengths: - The authors provide good theoretical justification of DivE2. In Section 3.2, the authors show that Algorithm 1 can converge to a local optimum for beta-strongly convex losses for a given (k,p,gamma,lambda) setting. The authors also describe how this approximate solution changes as (k,p,gamma,lambda) change over the course of the T episodes. - The diversity constraints and regularizers are all fairly interpretable by users. Users can directly tune the regularization terms and set the diversity constraints. - In experiments, the authors also combine DivE2 with multiple state-of-the-art dynamic classifier selection (DCS) methods. This makes sense, since training locally diverse base classifiers naturally lends itself to performing well with DCS methods. However, the authors only present test accuracies for these methods, while the main purpose of DCS methods is to speed up evaluation time. As an extension to these experiments (perhaps in a later paper), I would be interested to see experimental results for evaluation time speedups using DCS methods with DivE2. Specifically I’d be interested in the question, does combining DivE2 with DCS methods provide better evaluation time speedups than DCS methods alone? Weaknesses: - The DivE2 algorithm requires 4 user specified hyperparameters: T (the number of training “episodes”, where diversity requirements change for each episode), delta_k (change in k after each episode), delta_p (change in p after each episode), mu (change in gamma and delta after each episode. These hyperparameters all seem hard to select a priori, and would presumably have to be validated during training. - While DivE2 can reduce the number of examples seen by each base model, it also introduces additional overhead through the constrainted combinatorial optimization step. The authors show experimentally in Table 1 that the SubmodularMax step takes <10% of the total training time, which seems reasonable, so this is not a huge weakness. Recommendation: Overall, this paper presents a method for training diverse ensembles with strong theoretical and empirical results. My recommendation is to accept.